# Changes in the Current Patterns of Beef Consumption and Consumer Behavior Trends—Cross-Cultural Study Brazil-Spain-Turkey

**DOI:** 10.3390/foods12030475

**Published:** 2023-01-19

**Authors:** Danielle Rodrigues Magalhaes, Cihan Çakmakçı, María del Mar Campo, Yusuf Çakmakçı, Fausto Makishi, Vivian Lara dos Santos Silva, Marco Antonio Trindade

**Affiliations:** 1Department of Food Engineering, University of Sao Paulo, Av. Duque de Caxias Norte 225, Pirassununga 13635-900, SP, Brazil; 2Department of Agricultural Biotechnology, Animal Biotechnology Section, Faculty of Agriculture, Van Yüzüncü Yıl University, 65090 Tuşba, Van, Turkey; 3Department Animal Husbandry and Food Science, Instituto Agroalimentario IA2, Universidad de Zaragoza-CITA, Miguel Servet 177, 50013 Zaragoza, Spain; 4Department of Agricultural Economics, Faculty of Agriculture, Tekirdağ Namık Kemal University, 59030 Süleymanpaşa, Tekirdağ, Turkey; 5Department of Food Engineering—Institute of Agricultural Sciences, Federal University of Minas Gerais, Av. Universitaria 1000, Montes Claros 39404-547, MG, Brazil

**Keywords:** red meat, consumption habits, purchase decision, extrinsic factors, beef reduction

## Abstract

This cross-cultural study aimed to determine the main factors behind potential changes in eating habits by analyzing changes in the patterns of beef consumption currently observed in Brazil, Spain, and Turkey. To achieve this aim, 412 regular beef consumers from Brazil, 407 from Spain, and 424 from Turkey answered a self-administered questionnaire. The study surveyed the effects of economic factors, switching from beef to other sources of protein, aspects of credence, health-related concerns, the influence of lifestyle on beef consumption patterns, and purchasing decision factors. The most important factors that changed consumer behavior and resulted in a decrease in consumption, mostly among Brazilian and Turkish consumers, were the economics and accessibility of the products. Beef was replaced by other alternative sources of protein that were likewise derived from animals. The consumers whose purchasing intentions were most significantly influenced by credence factors (e.g., indiscriminate use of agricultural products, substandard animal welfare requirements, among others) were Brazilian and Turkish and, to a lesser degree, Spanish consumers. Lifestyle factors (e.g., consumption of out-of-home meals, available time to cook, among others) were demonstrated to alter consumption patterns and therefore must be carefully considered by the industry, taking into account cultural differences and consumer needs. The population under investigation considered that eating beef had no impact on their health.

## 1. Introduction

Considering that beef is an expensive protein source, the world’s consumption of beef continues to rise, particularly in emerging nations. With the result of emerging nations’ economic growth, it is now possible for more people to purchase and enjoy beef products. Additionally, structural factors including increased urbanization, employment creation, and spending on out-of-home food services have increased per capita beef consumption because of demographic change-causing growth [1]. On the contrary, consumers in developed nations are consuming less beef because the market is saturated, and they are more concerned about the environment, ethics, animal welfare, health consciousness, so they are also looking for alternatives to traditional animal proteins. Despite this decrease in beef consumption, developed countries continue to be the largest consumers in per capita proportion of meat in the world [2,3,4].

Although the reasons for choosing beef are the same across all consumers studied, their relevance differs across countries and across time [5,6,7,8]. It has been demonstrated that major factors of consumer preferences and changes in beef consumption behavior include the consumer’s environment and a greater understanding of the nutritional value of meat [9,10]. Consumer interest in assessing the economic, technological, social, and political implications of beef production and consumption is currently expanding. This entails choice based on considerations other than the typical intrinsic factors, which include appearance, texture, flavor, and odor [11,12,13,14].

The consumption of any product, including meat, is explained by the level of income and the changes in purchasing power that this level generates [7,15]. One reason why there is less demand for beef compared to other animal products such chicken and pork or plant-based foods such legumes, for example, is due to the lower cost compared to beef. Moreover, global economic problems, such as the COVID-19 pandemic and the ongoing conflict between Russia and Ukraine, show that economic factors must contribute to the drop in purchasing power even in developed countries, which are linked to high prices of goods. Imports and government restrictions on transport and trade can obstruct food production and supply chains around the world. However, despite having enormous importance by themselves, price elasticity and income elasticity are not enough to explain changes in consumer purchasing behavior [16,17,18].

Increasingly, consumers have the option to select beef based on a number of factors that are in accordance with credence quality (those product attributes that are difficult to assess even after consumption), which can provide significant consumer guarantees, such as certifications of origin, quality, sustainability, and others [19,20,21,22,23]. Concern for animal welfare, sustainability, and the prevention/control of disease outbreaks are other factors that have an impact on consumption. Human health hazards linked to the overuse of chemicals in agriculture and drug misuse in livestock further increase the perception of strict regulation for food safety [22,23].

The adoption of public policies along with incentives for the adoption of new technologies, such as those that increase animal productivity or limit the growth of animal numbers, thereby reducing pressure on natural resources, has shown that, when compared to previous periods, current meat production systems and the problems associated with them are being treated with higher responsibility [1,4,24]. The outcomes of these political and technological actions—a production process that is safe and respectful to the animal and the environment—can positively affect consumers’ intentions to buy or consume beef, just as their absence or faulty implementation have the opposite of the desired effect [18,21,25,26,27,28,29].

Consumers’ health concerns also have a significant impact on their intentions to consume beef [30,31]. The intake of beef is linked to the fact that it offers significant health benefits because of its nutritional value as a source of high-quality protein, Fe, Zn, and vitamin B12. As a result, beef consumption and its effects on health has been a topic of intense debate. On the other hand, research indicates that consuming large amounts of beef may increase the risk of developing chronic illnesses such diabetes, cancer, cardiovascular disease, and high rates of death. From these studies, beef is viewed as both healthy and unhealthy, which causes conflict in the minds of consumers [8,11,32,33,34,35,36,37]

Regarding a population’s social composition, there are a number of well-known variables that affect the consumption of beef, including gender, ethnicity, socioeconomic status, and level of education, among others. The intake of beef can be affected by lifestyle factors, which are closely related to those sociodemographic characteristics [36,38]. Changes in family structure alter the volume of beef intake, where the rising participation of women in the workforce, which results in changes in the convenience of food preparation and purchase due to the lack of time for cooking as well as the average family size, being reduced. The social environment in which consumers interact, experience, and learn about the nutritional benefits and environmental impacts includes access to social media that inspires consumers to prepare gourmet recipes and can also influence habits, for example, domestic and extra-domestic consumption. Regional aspects include food requirements, availability of products, local or convenient shopping, traditions, and even the weather seasons [2,4,7,11,24,39,40].

As shown, the ethical environmental aspects of beef production, the concern for health and lifestyle, and, of course, economic considerations can all be seen as valid factors that affect changes in consumption patterns and consumer behavior that can affect eating habits. For the beef chain to fully comprehend the situation, make decisions, and assess the mechanisms influencing consumer choice, research on consumer behavior must be studied and updated [2,6,8,27,40,41].

Therefore, this study aimed to cross-culturally analyze changes in current beef consumption patterns in a Brazilian, Spanish, and Turkish population, with the aim of finding the main motivations responsible for possible changes in eating habits.

## 2. Materials and Methods

### 2.1. Sample—Consumers and Study Location

This study performed the application of a self-administered questionnaire during the second semester of 2021, collecting a total of 1243 complete and valid responses from regular beef consumers, including 412 responses from Brazil, 407 from Spain, and 424 from Turkey. The questionnaire was created using the Google Forms software (Web Application—Google Platform) and was sent online to consumers in the native language of each country: Portuguese, Spanish, and Turkish.

The analysis was planned to achieve descriptive and empirical goals using two non-probabilistic sampling techniques: conventional sampling, in which individuals were chosen for their accessibility, and snowball sampling, which was used to access with specified criteria, in this case, beef consumers [42,43]. Researchers from Sao Paulo University in Brazil, Zaragoza University in Spain, and Yuzuncu Yıl University in Turkey sent the questionnaire to their own contacts by email, media platforms, and also through posters with a “QR code” (quick response code) placed at well-known establishments for the respondents to access them. After completing the questionnaire, the respondents were invited to disseminate it to their connections who consumed beef, thereby increasing the number of people covered. The questionnaire remained available online until a minimum of 400 valid responses per country was reached, a number that was determined based on the literature [12].

The questionnaire included questions about the frequency of beef consumption (Table 1), sociodemographic characteristics (Table 2) and six closed-ended questions, shown below exactly as they were phrased in the survey, with two or more possible answers about the factors that impact changes in beef consumption and also consumer behavioral trends.

Has your beef purchasing pattern changed due to economic reasons in the last two years?Have you started using other sources of protein in your diet with the intention of replacing beef in the last two years?Which protein sources below have you consumed to replace beef in your diet in the last two years?Have any of these factors (credence) affected your level of credence and, as a consequence, the beef purchase intention?Has your beef consumption changed because of those health-related factors? (The factors were described below the question).Due to the following (lifestyle) factors: Have you increased, decreased, or not changed your beef consumption?

Due to the various levels of socioeconomic development, consumption patterns, and cultural aspects that affect consumer behavior patterns, an intercultural study allows for the highlighting of different beef consumption and production scenarios. Brazil is a country that stands out internationally for its significant contribution to the production of beef. As one of the main agricultural commodities, exporting beef is crucial to the country’s economy [44]. Brazilians are the third-largest consumer of beef in the world (24.6 kg/person/per year) thanks to easy access to meat, a variety of cuts, and traditions surrounding its consumption during special occasions [1,7]. On the other hand, Spain is the fifth-largest beef exporter in the European Union [45]. The demand for beef (12.1 kg/person/per year) shows a seasonal pattern, being higher in the winter and noticeably lower in the summer [46]. Recent years have seen a decrease in the consumption of beef in Spain as well as other developed countries [47,48]. Finally, Turkey imports most of the beef consumed due to a deficit in the livestock industry, which is caused by issues including family production and inefficient native breeds [48]. One of the main reasons beef consumption is not higher is due to the expensive red meat on the market, where the current consumption is 11.1 kg/person/per year [1,49].

### 2.2. Data Analysis

The program SPSS version 28 (IBM^®^ Statistics, SPSS Inc., Chicago, IL, USA) was used to perform the statistical analysis. A descriptive analysis of the data was conducted to identify the factors investigated among countries, using cross tables to determine the frequency of attributes when Pearson’s chi-square was less than 5% and the z-test adjusting *p*-value by the Bonferroni technique to establish comparisons. The variances between the consumer mean values in the three countries were compared using the ANOVA test, which was likewise performed with a significance level of ≤0.05.

### 2.3. Experimental Overview

The frequency of beef intake among Brazilian, Spanish, and Turkish consumers is shown in Table 1. The majority of Brazilians are consumers of beef who eat it twice weekly (72.6%). Consumers from Spain and Turkey consume on average once a week (42.8% and 43.4%, respectively). Turkish consumers (16.7%) have the lowest frequency of once a month or less when compared to other consumers.

Furthermore, socioeconomic data were collected and are presented in the table below (Table 2). In summary, contrasted to Turkey, Brazil and Spain surveyed more female individuals. Younger consumers, those between the ages of 18 and 34, were more prevalent across all countries. Most of those questioned from the three countries had a high level of education although their monthly incomes varied. Three or more persons formed the family environment. A sample of people who eat beef in each country was used for the study; it should be highlighted that this sample is not representative of the population as a whole.

## 3. Results and Discussion

### 3.1. Influence of Economic Factors in Purchasing Decisions

The first question of this study explores whether the buying pattern of beef was modified due to economic factors in the years 2020–2021, during which the price of beef increased in response to the global crisis caused by the COVID-19 pandemic.

The findings in Table 3 show that 64.6% of Brazilian consumers reported reducing regular beef purchases due to high prices, followed by Turkish consumers (55%) and only 11.1% of Spanish consumers. Contrarily, the majority of consumers in Spain (87.5%) answered that economic factors had no impact on their consumption habits for beef, followed by Turkish (43.4%) and Brazilian (33.5%) ones. Less than 2% of consumers across the three countries indicated to have increased their meat purchases as a result of their increased purchasing power.

The economic aspect is one of the classic factors that explain changes in consumption habits [6,7,15]. Even though Brazil is one of the world’s leading producers of beef, over the past two years, mainly as a result of rising agricultural input costs and the increase in production that is focused on exports, the country ended up inflating the domestic market, increasing the price of beef by 42.6% between March 2020 and April 2021 [50]. In this study, it was observed how an economic issue can have a great impact on the consumption of beef; however, the economic factor has been for some years a factor of decrease in consumption, as exemplified by the results of a 2016 study with consumers of beef from Brazil and Spain that found that the majority of the survey participants, who were frequent consumers of beef, had not changed their consumption levels in the previous years [12].

The accessibility and relatively low cost, when compared to most other locations around the world, are the primary reasons for the high consumption in Brazil [11,12]. Consequently, due to the recent economic issues, the majority of consumers reported experiencing a reduction in beef consumption during the past two years.

On the other hand, Spanish consumers managed to maintain stable beef consumption. According to meat consumption study, wealth and price will have less of an impact over time, and many markets may have reached saturation in terms of meat consumption. As a result, other elements such as quality will play a bigger role in influencing consumer decision making. These results support the findings for Spanish consumers in our study. Despite the fact that the consumption trend for beef is, in fact, expected to reduce gradually, Spain is classified as an economically stable country [47], has greater purchasing power, and has demonstrated the ability to support recent economic changes more than the other two countries under study.

Turkey, a country that imports beef, was affected by the pandemic’s economic crisis, which included higher prices, a scarcity of products, and, consequently, significant reductions in consumption [51]. Researchers also have demonstrated that the influence of the factors that affect meat consumption is not uniform across different types of meat by utilizing the example of beef. In contrast to the pattern for meat as a whole, it reveals a significant decline in the consumption of beef in many parts of the world. The cost of beef in comparison to another protein source, such as chicken, is a significant factor in this difference.

However, the percentage of consumers who started consuming less beef was larger in both countries than the percentage who began consuming more. The increase in price was the main factor identified for the decline in consumption [12]. According to consumer responses from Brazil and Turkey, it was found in the current study that less beef was being purchased as a result of increased costs.

### 3.2. Replacing Beef with Other Sources of Protein

In consideration of the introduction of other sources of protein to the diet, this topic investigated how beef consumption patterns have been modified. The findings are presented in Table 4 and demonstrate that there are statistical differences among consumers in the three countries with regard to the factors analyzed (*p* ≤ 0.001).

Compared to 35.9% of Brazilians and 17.1% of Turks, 61.5% of Spanish consumers did not introduce another source of protein with the intention of replacing beef. The introduction of other sources of protein into their diets led to a decrease in beef consumption according to a larger percentage of Brazilian consumers (53.2%), followed by Turkish consumers (38.7%). On the other hand, even with the introduction of other sources of protein to the diet, a 24.2% increase in beef consumption was reported by Turkish consumers, which is twice as high as the percentages reported by beef consumers in Brazil (11%) and Spain (11.7%).

Additionally, it was studied whether beef was replaced in the consumers’ diets with other meat and vegetable products rich in protein (beans, chickpeas, lentils, tofu, among others) (Table 5). The consumers who answered the previous question by saying they had begun to consume less beef as a result of the introduction of other sources of protein were those who expressed their opinions on the subject.

According to the results, Brazilians most frequently substituted beef for pork (49.6%), chicken (46.4%), and plant-based protein sources (30.4%). Spanish consumers choose to substitute beef for vegetal protein sources (35.5%) and lamb/goat meat (33.1%) in their diets. Turkish consumers choose vegetable protein sources (34.1%) and lamb/goat meat (47%) in place of beef.

Contrary to the other meat sources analyzed, where significant differences (*p* ≤ 0.001) were found among countries, replacing beef with plant-based protein did not result in any significant variations among countries (*p* = 0.193). Brazilians typically substituted more beef for pork and chicken, whereas Turks usually substitute for more lamb or goat meat.

Most of the Turkish consumers questioned had incomes that were higher than those of the Turkish population as a whole (Table 2). Due to high income, they manage to maintain or even increase the consumption of beef even with the introduction of other protein sources in the diet. This helped to define the habits of the beef-consuming population for this study; however, it undoubtedly does not explain Turkish consumers as a whole. In the other two investigated countries, this was not detected.

Our findings are in line with the Organisation for Economic Co-operation and Development—OECD (2022) [1], which states that the demand for meat does not generally change much with regard to quantity. What takes place is the variance in the sources used to compose diets; low-cost protein sources have taken the place of beef [52]. The literature supports the Brazilian and Turkish findings, which demonstrate that when economic issues impact emerging economies, the type of protein consumption alters [41,53,54].

According to previous research, it is possible that the accessibility of products in each country contributed to the movement in consumer preference from beef to other sources of protein under investigation [12,40]. The largest substitution in Brazil occurred from the substitution of beef for other meat products, and during the study period, Brazil had the most affordable prices and a good supply of pork and chicken [50].

Spanish consumers, compared to the studied Brazilian and Turkish consumers, reported the lowest percentage of change in beef consumption (61.5%—Table 4); only 26.5% answered they ate less beef due to the introduction of meat from lamb/goats and protein-rich vegetable. The research indicates that changing from meat consumption to plant-based products, especially legumes, which are strong sources of protein, iron, and zinc, can result in considerable environmental impact reductions without sacrificing the health benefits of meat [37]. As meat is regarded as a source of pleasure, this would necessitate that beef consumers be aware of and interested in these meat substitutes as other protein sources [7,30].

Turkey is a significant producer of lamb and goat meat [51], and due to availability, Turkish consumers replaced beef with lamb or goat meat. According to other studies, lamb is an equally costly protein as beef when it comes to other red meats such as beef, so the decision to change can be explained by factors such as personal satisfaction, convenience, or availability [55,56]. Likewise, findings demonstrate that substituting vegetable proteins for animal proteins is a more efficient way to transition to a more balanced diet of animal proteins in countries with a high meat consumption. An alternative for consumers who prefer to eat less beef is to replace beef consumption with other red meat in particular (pork and lamb) or to substitute beef for chicken meat, which contributes to reducing environmental impact [8,21,31,37].

### 3.3. Importance of Credence in the Purchase Process

The following factors described in Table 6 were studied: animal welfare, environmental aggression, indiscriminate use of agricultural products, food adulteration/contamination, and loss of trust in production systems as a result of the crisis caused by the COVID-19 outbreak.

Brazilians, in general, were the consumers who saw their purchase intention most strongly influenced by the credence factors investigated in this subject.

A total of 21.6% of the Spanish consumers studied affirmed that animal welfare and 23.6% confirmed that the indiscriminate use of agricultural products are factors that would influence the purchase decision. The criteria that would be taken into account when evaluating the purchase intention in the Turkish population under study are the indiscriminate use of agricultural products and the environmental impact caused by animal production (32.9% and 32.8%, respectively).

Brazilians’ credence was impacted by the COVID-19 pandemic’s concerns in the meat system, as expressed by 77.7%, although Spaniards’ and Turks’ credence was minimally impacted (12.1% and 10.2%, respectively). Brazilians may have been uncertain about the product’s reliability due to high beef prices and concerns about internal shortages.

Consumers’ preferences to purchase beef may also be influenced by factors that affect their level of credence in the animal production system [8,12,26,27,35,57]. A study of Brazilian consumers showed that Brazil’s status as a big beef producer pushes the population closer to agricultural news, increasing their chance of knowing about corruption and poor technical–political rules used in the production system [7]. This may help to explain why Brazilian consumers’ purchasing intentions appear to be severely impacted by their level of credence in aspects of the animal production system.

In any case, because these factors are subjective, consumers cannot make decisions about meat without quality certificates that support these credible characteristics on meat labels [5,6,9,25,55].

Considering the evidence presented in the literature in which concerns about animal welfare and environmental impact are the most frequent justifications for avoiding meat, it would be reasonable to assume that placing less emphasis on ethical considerations in food selection would encourage consumers to consume less beef or to avoid it altogether [18,24,58]. Therefore, many researchers suggest decreasing consumption in favor of reducing the environmental impact caused by the meat sector [18,24,58], while others encourage good production techniques since animal welfare standards can be improved, the health benefits of meat can be increased, and harmful effects on human health and the environment can be reduced through innovation and behavioral change [17,27,30,59].

### 3.4. Changes in Beef Consumption due to Health Reasons

This topic aims to establish the link between changes in beef consumption patterns as a result of health factors, such as the need to prevent or control diseases or maintain one’s personal health.

According to the result shown in Table 7, 80% of the beef consumers in the three countries did not change their beef consumption patterns as a result of health concerns either in an effort to prevent or control them. The results of this study represent young consumers, and they might have been different if the population under study had been older.

Consumers in Brazil (9.7%), Spain (14.3%), and Turkey (12.7%) started consuming less beef to avoid or treat diseases. Nevertheless, a small percentage of consumers (8.3% in Brazil, 7.1% in Spain, and 5.0% in Turkey) increased their beef consumption on the premise of maintaining their health.

There were no significant differences among consumers in the three countries under study (*p* = 0.111).

Many studies are inclined to recommend the reduction of red meat consumption due to its effects on health, where they claim that excessive consumption of meat has been linked to chronic diseases, cancers, weight gain, and other conditions. These studies come from a variety of sources, including medical institutes, universities, and non-governmental organizations [56,57,58,59,60]. Red and processed meat consumption should be maintained at recommended levels as part of a healthy and environmentally friendly diet [8,30,40]. Excessive meat consumption, which is linked to the lowest food quality observed, supports initiatives and policies that encourage this decrease.

In a study with young adult beef consumers, the findings regarding the motivations for reducing beef consumption revealed that those who are in favor of doing so (22.3% of the participants) cite ethical and health concerns, while those who are strongly opposed to reducing (42%) mention pleasure and diet quality. Researchers suggest a correlation between changes in beef intake behavior and a preference for choosing cuts with reduced fat content [7,55,61]. The reduction or even discontinuation of beef consumption comes from medical advice to individuals with chronic health problems and not to healthy beef consumers [33,56,60].

Similar to this, research in the literature indicates that concerns about one’s health are the key factors in people choosing to eat less beef. However, our findings show that beef eaters did not reduce their beef consumption for health reasons.

### 3.5. Lifestyles as a Factor for Changes in Beef Consumption Habits

Therefore, the objective of this final topic was to determine whether recent changes in family lifestyle had increased, decreased, or had no effect on beef consumption. Table 8 provides the findings in relation to this study.

The results demonstrate that most consumers in the three countries under study did not change their consumption as a result of factors related to recent changes in lifestyle. Despite this, a significant number of respondents indicate having changed their consumption in some way.

Analyzing each of the factors, it was found that the number of family members decreased the percentage of beef consumption in Brazil (25.7%), Spain (19.7%), and Turkey (15.6%). When children and elderly family members are present, beef consumption rises, most notably in Turkey (21.2%) and less in Spain (16.7%) and Brazil (15.3%).

Brazil (25%) and Spain (13.5%) saw an increase in consumption due to convenience, whereas Turkey (26.7%) saw a reduction. For 32.3% of Brazilians, the time available for cooking resulted in a reduction in beef consumption, followed by consumers in Spain (17.0%) and Turkey (13.7%). Brazilians (25.2%) and Spanish consumers (21.1%) followed by Turkish consumers (31.0%) increased their beef intake due to increased extra domestic consumption. The gourmetization of foods increases the consumption of beef, especially in Brazil (29.1%) and Turkey (25%) but less in Spain (11.3%).

Finally, 14.5% of Spanish consumers, 18.2% of Turks, and 22.8% of Brazilians reported that they started eating more beef as a result of their increased physical activities.

Lifestyle habits, which are closely connected to socioeconomic and demographic factors, can change how much beef is consumed [7,10,62].

The number of family members, the presence of children and elderly people in the family environment, convenience (easy access to places of purchase), time available for cooking, extra domestic consumption, gourmetization (refers to the processes or trans-forming food products from simple into exclusive products), and physical activities are some topics found in the literature that may be able to change the consumption of beef [4,40,41,62]. Consumers have historically relied significantly on intrinsic cues to make judgments about the quality experience of beef. However, modern consumers seek to meet their everyday demands and anticipate the experience’s quality to match their expectations influenced by lifestyle factors [63,64].

It is observed that Brazil is currently experiencing greater changes in lifestyle due to a decrease in the number of family members and greater inclusion of women in the labor market [4,9]. In addition, social media is thought to have a huge influence on how often people eat out, prepare more gourmet food at home, and how much they exercise, which is causing the increase in beef consumption [65,66].

Some differences, such as the presence of children but primarily elderly people who live in the family environment and not alone, are relevant from a cultural point of view with regards to the Turkish population [67]. Additionally, it is important to note the amount of extra-domestic consumption because of the lower costs of eating out than in Brazil and Spain. On the other hand, Brazil and Spain have a large number of supermarkets with butcher shops within, which may be more convenient and raise consumption there [12], which is not the case in Turkey, showing that limited accessibility could be the reason for the reduction in consumption.

### 3.6. Limitations

The results of this study, which focused on beef consumers and cannot be extrapolated to individuals who do not purchase or consume beef, are consequently not representative of the populations of the three countries under investigation. The potential for self-selection bias and the limited ability for personal contact are additional limitations, as is the case with all voluntary and online surveys, which increases the possibility of fraudulent responses.

## 4. Conclusions

This study attempted to comprehend how beef consumption patterns among Brazilians, Spanish, and Turkish consumers have changed in the past two years and explore the factors that have contributed to these changes.

Our primary purpose was not to evaluate these effects during the COVID-19 pandemic, but since the study period coincided with the pandemic period, some factors that were studied were amplified, such as economic and accessibility factors, which have contributed to our better understanding of how fragile the food production sectors are that affect the meat production chain and, as a result, the consumer’s purchasing power and consumption intentions.

The findings also revealed that about one-third of consumers prefer replacing beef with vegetable proteins, whereas more Brazilians and Turks opted to substitute beef with other animal proteins. Due to the socio-environmental aspects of animal production, the results of the crises in the meat industry, and other factors that have an impact on consumer credence, a significant percentage of consumers are reducing their consumption of beef. It is also highlighted that in order for industries to more effectively satisfy consumer demands, lifestyle factors that drive changes in consumption habits must be carefully considered.

The effects of beef consumption on health were evaluated in young adult, presumably healthy consumers, and the findings indicate that this segment of the population did not alter their consumption as a result of health conditions or for the prevention of those conditions.

## Figures and Tables

**Table 1 foods-12-00475-t001:** Frequency of beef intake among consumers in Brazil, Spain, and Turkey.

Frequency	Country			*p*
	Brazil	Spain	Turkey	
Twice a week or more	72.6 a	27.3 b	30.2 b	≤0.001
Once a week	15.3 b	42.8 a	43.4 a	
Once every 15 days	3.6 c	20.1 a	9.7 b	
Once monthly or less	8.5 b	9.8 b	16.7 a	

Each letter indicates a subset of categories (country) whose column proportions do not differ significantly from each other at the 0.05 level.

**Table 2 foods-12-00475-t002:** Frequency of sociodemographic data in percent for gender, age, educational level, income, and family members of consumers studied in Brazil, Spain, and Turkey.

Socioeconomic Data		Country		
		Brazil	Spain	Turkey
Gender	Man	41.3	32.4	67.2
	Woman	58.7	67.6	32.8
Age	18 to 34 years old	40.7	31.5	38.2
	35 to 44 years old	16.5	19.7	28.1
	45 to 54 years old	29.5	25.3	20.8
	Over 55 years of age	13.3	19.9	13.0
Educational Stage	Primary	2.6	1.0	0.5
	Secondary	9.0	4.7	5.2
	Technician course	6.2	11.5	12.3
	University	82.2	82.8	82.0
Monthly Income *	Up to two minimum salaries	19.6	60.7	14.3
	Between 2 to 4 salaries	22.8	14.5	32.8
	Between 4 to 6 salaries	17.7	8.6	28.5
	Between 6 to 9 salaries	11.9	1.0	10.4
	More than 9 salaries	19.2	15.2	7.1
	Not informed	8.7	0.0	5.9
Household members	Alone	8.7	9.8	3.0
	2 people	25.0	21.4	17.7
	More than 3 people	66.3	68.8	79.3

* Family income was asked according to the currency of each country. For comparison purposes, quoted on 08/04/2022, the conversion from real (Brazil; BRL) to euro (Spain; EUR) was (1:5.35) and from real to Turkish lira (Turkey; TRY) from (1:0.29). Sampling: Brazil (*n* = 412), Spain (*n* = 407), and Turkey (*n* = 424).

**Table 3 foods-12-00475-t003:** Changes in beef purchase pattern due to economic aspects.

Changes in Beef Purchase Pattern due to Economic Aspects	Country			*p*
	Brazil	Spain	Turkey	
Reduced due to the high price of beef	64.6 a	11.1 c	55.0 b	≤0.001
Increased due to increased family income	1.9 a	1.5 a	1.7 a	
No change due to economic reasons	33.5 c	87.5 a	43.4 b	

Each letter indicates a subset of categories (country) whose column proportions do not differ significantly from each other at the 0.05 level. Sampling: Brazil (*n* = 412), Spain (*n* = 407), and Turkey (*n* = 424).

**Table 4 foods-12-00475-t004:** Change in beef consumption habits due to the introduction of other sources of protein in the diet.

Change in Beef Consumption by the Introduction of Other Protein Sources	Country			*p*
	Brazil	Spain	Turkey	
There was no introduction of other protein sources intended to replace beef	35.9 b	61.5 a	17.1 c	≤0.001
Less beef is consumed due to the introduction of other protein sources	53.2 a	26.8 c	38.7 b	
More beef is consumed even with the introduction of other protein sources	11.0 b	11.7 b	24.2 a	

Each letter indicates a subset of categories (country) whose column proportions do not differ significantly from each other at the 0.05 level. Sampling: Brazil (*n* = 412), Spain (*n* = 407), and Turkey (*n* = 424).

**Table 5 foods-12-00475-t005:** Introduction of other sources of protein to replace beef in the Brazilian, Spanish, and Turkish population questioned.

Introduction of Other Dietary Protein Sources in Recent Two Years	Country			*p*
	Brazil	Spain	Turkey	
Pork	49.6 a	17.0 b	-*	≤ 0.001
Chicken	46.4 a	26.9 b	26.7 b	≤ 0.001
Lamb/goat meat	19.9 c	33.1 b	47.0 a	≤ 0.001
Plant-based	30.4 a	35.5 a	34.1 a	0.193

Each letter indicates a subset of categories (factors) whose column proportions do not differ significantly from each other at the 0.05 level. Sampling: *n* = Brazil (219), Spain (109), and Turkey (164). * For reasons of predominantly Muslim population, pork consumption was not asked.

**Table 6 foods-12-00475-t006:** Credence factors in beef production system that negatively affect the purchase intention of beef of Brazilian, Spanish, and Turkish consumers.

Factors Affecting the Level of Credence in the Beef Production System	Country			*p*
	Brazil	Spain	Turkey	
Animal welfare requirements not fulfilled	52.9 a	21.6 b	25.5 b	≤0.001
Environment impact caused by animal production	57.5 a	9.7 c	32.8 b	≤0.001
Indiscriminate use of agricultural products	55.4 a	23.6 b	21.0 b	≤0.001
Adulteration/contamination of the product	49.6 a	17.5 c	32.9 b	≤0.001
Uncertainties created by the COVID-19 pandemic	77.7 a	12.1 b	10.2 b	≤0.001

Each letter indicates a subset of categories (factors) whose column proportions do not differ significantly from each other at the 0.05 level. Sampling: *n*= Brazil (412), Spain (407), and Turkey (424).

**Table 7 foods-12-00475-t007:** Changes in beef consumption due to health factors.

Changes in Beef Consumption due to Health Factors	Country			*p*
	Brazil	Spain	Turkey	
The beef consumption was not affected by health problems	82.0 a	78.6 a	82.3 a	0.111
Increased consumption—maintain health	8.3 a	7.1 a	5.0 a	
Decreased consumption—disease prevention	9.7 a	14.3 a	12.7 a	

Each letter indicates a subset of categories (country) whose column proportions do not differ significantly from each other at the 0.05 level. Sampling: Brazil (*n* = 412), Spain (*n* = 407), and Turkey (*n* = 424).

**Table 8 foods-12-00475-t008:** Factors related to lifestyle that modify beef consumption by consumers in Brazil, Spain, and Turkey.

Everyday Factors that Can Modify Beef Consumption		Country			*p*
		Brazil	Spain	Turkey	
Number of people in the household	Increase	17.2 a	12.8 a	12.7 a	≤0.001
	Decrease	25.7 a	19.7 a,b	15.6 b	
	No changes	57.0 b	67.6 a	71.7 a	
Presence of children and elderly people in the family environment	Increase	15.3 b	16.7 b	21.2 a	0.004
	Decrease	13.6 a	7.1 b	9.2 a,b	
	No changes	71.1 a	76.2 a	69.6 b	
Convenience/ease of buying beef	Increase	25.0 a	13.5 b	11.6 b	≤0.001
	Decrease	18.9 b	9.3 c	26,7 a	
	No changes	56.1 b	77.1 a	61.8 b	
Available time to cook	Increase	7.5 a,b	8.1 a	4.0 b	≤0.001
	Decrease	32.3 a	17.0 b	13.7 b	
	No changes	60.2 c	74.9 b	82.3 a	
Consumption of out-of-home meals	Increase	25.2 a,b	21.1 b	31.1 a	≤0.001
	Decrease	12.4 a	12.0 a	14.2 a	
	No changes	62.4 a,b	66.8 a	54.7 b	
Gourmetization	Increase	29.1 a	11.3 b	25.0 a	≤0.001
	Decrease	7.5 a	11.3 a	9.4 a	
	No changes	63.3 b	77.4 a	65.6 b	
Due to physical activities	Increase	22.8 a	14.5 b	18.2 a,b	≤0.001
	Decrease	12.9 a	7.1 b	8.5 a,b	
	No changes	64.3 b	78.4 a	73.3 a,b	

Each letter indicates a subset of categories (country) whose column proportions do not differ significantly from each other at the 0.05 level. Sampling: Brazil (*n* = 412), Spain (*n* = 407), and Turkey (*n* = 424).

## Data Availability

The data presented in this study are available on request from the corresponding author.

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
