# Peer review of "Changes in the Current Patterns of Beef Consumption and Consumer Behavior Trends—Cross-Cultural Study Brazil-Spain-Turkey"

_foods, 2023, doi:10.3390/foods12030475_

Round 1
Reviewer 1 Report
The paper presents the results of a cross-cultural study aimed at determining the main determinants of potential changes in dietary habits. The authors examined changes in the patterns of beef consumption in Brazil, Spain, and Turkey among 412, 407 and 424 respondents, respectively. A self-administered questionnaire was employed. The results indicate that economics and accessibility were the most important factors that changed consumer behaviour and decreased beef consumption, mostly among Brazilian and Turkish respondents.
The topic is highly relevant for various disciplines and has numerous policy implications. However, the paper has several serious shortcomings which need to be addressed:
1.
The main issue is the readability of the paper. Although grammatically, sentences and paragraphs are, for a large part, appropriate, they are often difficult to understand what the authors wanted to comminucte. Paragraphs are often too short, not following the scientific paragraph structure, so the main idea of a paragraph is often not developed. The paragraphs are sometimes only loosely linked together, jumping from one topic to another. This makes the job of a reader very difficult.
Just one example from the first paragraph:
“Although the relevance of the selection criteria for beef can vary over time and 35 among nations, they are generally the same.”
Selection criteria for whom?
And the second paragraph:
“Assessing the quality of beef can be more challenging than assessing the quality of 40 other consumer products.”
Assessed by who? Second, is it more difficult than, for example, pork? If so, why?
2.
Besides general readability, there are grammatical issues as well, for example:
“The con- 26 sumers whose purchasing intentions were most significantly influenced by confidence factors were 27 Brazilians e Turkish, and to a minor extent, Spanish consumers. Lifestyle has been demonstrated to 28 alter consumption patterns,”
“Brazilians e Turkish” needs correction.
What are confidence factors? When mentioned for the first time like this, the reader won’t understand it.
“Lifestyle has been demonstrated to 28 alter consumption patterns, and, therefore, must be carefully considered by the industry, taking into 29 account cultural differences and consumer needs.”
What about lifestyle specifically? Unclear.
“According to the studied population, eating beef 30 has no negative impact on one's health.”
This is again unclear. If the authors wanted to says that the perception of respondents is that beef consumption has no impact on health, I suggest rephrasing it that way. Additionally, this needs to be followed with a sentence on the negative impact of red meat, in general, and beef, specifically, on health outcomes, with, preferably, three or more references confirming this link, which is well-established in the literature.
3.
“The sociodemographic characteristics of the region, such as food preferences, life- 48 style, the accessibility of products, traditions, and even the seasons, can all have an impact 49 on consumption and purchase habits [11-15]. 50”
This is true, but not further explained in any way. The theoretical part needs a Literature review on sociodemographic and other determinants of meat consumption.
4.
“The world's consumption of beef has been increasing as a result of the population's 51 rising food requirements. Economic expansion has led to this demand, which is also a 52 consequence of other structural changes including population growth and the urbaniza- 53 tion process, particularly in developing nations [16]. 54”
In what way do economic expansion and urbanization drive meat consumption?
5.
“Health care has been a topic of significant discussion, and this aspect may have con- 72 tributed to changes in consumption patterns. On the one hand, beef consumption is re- 73 lated to the fact that it provides great health benefits, such as the provision of iron and 74 crucial nutrients. On the other hand, researchers suggest that a high intake of beef has 75 been associated with chronic diseases such as diabetes, cancers, cardiovascular disease, 76 and high mortality rates. Because beef is both considered healthy and unhealthy at the 77 same time, this ambiguity creates conflict in the consumer's mind [23-26]. 78”
This is, unfortunately, very misleading for the reader. What reputable scientific research or public health institutions claim, based on empirical evidence, that increased beef consumption is healthy compared to less beef consumption? If any, a reference is needed. Second, “provision of iron and crucial nutrients” is insufficient to claim a product has beneficial health effects. Many products have iron and crucial nutrients, but the health impact of a particular food or food group involves whether these nutrients can be acquired in other products, which tend to have (more) beneficial (or less detrimental) health consequences. Third, as iron is mentioned, let me note that high iron stores have been found to be associated with an increased risk of metabolic syndrome, diabetes, cardiovascular diseases and some forms of cancer. Even in the case of greater bioavailability of animal-derived iron (heme-iron), it has been associated with adverse health outcomes compared to plant-based non-heme iron.
To sum up, authors should put more thought into the construction of their sentences when making health claims of specific products/food and align them with scientific evidence.
6.
“Research has been showing how, in comparison to past periods, today's meat pro- 86 duction systems and the problems associated with them are given more consideration 87 than they were [15,18].”
Unclear. Furthermore, the present-day meat production system is getting more mass-based, extensive and deleterious for the planet and health (and animals) than it was 50 years ago. The authors themselves state that meat consumption is on the rise globally.
7.
“The application of a self-administered questionnaire to a total of 1243 regular con- 114 sumers of beef in Brazil, Spain and Turkey was performed, for this study, during the sec- 115 ond semester of 2021. 116”
The sample description is too vague. It is not clear how representative (and representative of whom) the sample is in each of the three countries.
“The questionnaire, which was created with the help of the Google.forms software 137 (Web Application - Google Platform), was sent online to consumers in the native language 138 of each country: Portuguese for Brazil, Spanish for Spain, and Turkish for Turkey. 139 Data were collected using two non-probabilistic sampling techniques: conventional 140 sampling, in which subjects are chosen for their accessibility, and chain sampling, which 141 is used to categorize subjects with specified criteria, in this case, beef consumers [41,42].”
It is unclear how, specifically, data was collected with regard to the conventional sample (“subjects were chosen” – but according to what criteria? What was the response rate? The same goes for chain sampling.).
All this makes the reproducibility of the study practically impossible.
8.
“Aspects of confidence as a decisive factor in the purchase.“
Who’s confidence in what?
9.
“The influence of lifestyles in beef consumption pattern”
“In” à on. Pattern à Patterns.
10.
“In summary, contrasted to Turkey, Brazil and Spain surveyed more female in- 170 dividuals. Younger consumers, those between the ages of 18 and 34, were more prevalent 171 across all countries.”
ANOVA mean differences may primarily be due to the differences in sample characteristics among the three countries. This puts the validity of the results under serious doubt.
11.
“it should be highlighted that this sample is not representative of the population as 175 a whole.”
The “national” results may primarily be due to sample characteristics, while the readers don’t know much about data collection.
Sociodemographic differences between samples should be taken into account (empirically controlled for), and then mean differences can be discussed.
12.
The Results section:
“The economic aspect is one of the classic factors that explain changes in consumption 184 habits [2,3,19].”
Stating references need to be done in the Theoretical part and then in the Discussion section.
13.
“In this study it is observed how an economic crisis can have a great impact on the 215 consumption of beef, however, the economic factor has been for some years a factor of 216 decrease in consumption as exemplified by results of a 2016 study with consumers of beef 217 from Brazil and Spain found that the majority of the survey participants, who were fre- 218 quent consumers of beef, had not changed their consumption levels in the previous year’s 219 [32]. 220”
What is the specific evidence/measure that indicates that the economic crisis had a great impact on the consumption of beef?
14.
Table 4:
“There was no introduction of other protein sources intended to replace beef”
“Less beef is consumed due to the introduction of other protein sources.”
“More beef is consumed, even with the introduction of other protein sources.”
The first statement does not necessarily indicate that there was no change in meat consumption but merely that reduced meat was not replaced with other sources of protein.
15.
“Table 5. Introduction of other sources of protein to replace beef in the Brazilian, Spanish and Turkish 253 population questioned.”
What is the number of respondents in each country in this particular analysis? Not the full sample surveyed?
16.
3.5 “Lifestyle habits, which are closely connected to socioeconomic and demographic fac- 351 tors, can change how much beef is consumed [3,10,55]. 352”
A sociodemographic literature review on beef and meat consumption needs to be added to the theoretical part. The authors may take advantage of MDPI journals’ and other publications on determinants of food choice and include them for an interested reader, e.g.,:
1) https://doi.org/10.3390/su132313036
2) Gossard, M.H.; York, R. Social Structural Influences on Meat Consumption.Hum. Ecol. Rev.2003,10, 1–9.
3) Koch, F.; Heuer, T.; Krems, C.; Claupein, E. Meat consumers and non-meat consumers in Germany: A characterisation based on rresults of the German National Nutrition Survey II.J. Nutr. Sci.2019,8, e21
17.
“This study attempted to comprehend how beef consumption patterns among Brazil- 397 ians, Spanish and Turks consumers have changed over time”
What was the time period asked about?
18.
“The results also showed that the behavior of beef consumers was the preference for 406 replacing beef with other animal sources of protein.”
Unlcear.
19.
There is no Discussion section that would put results into context with broader literature.
The Limitations section/paragraph is missing.
I suggest the authors thoroughly revise the paper, as the topic of the paper is of the most importance to humans, animals and the planet.
Author Response
The paper presents the results of a cross-cultural study aimed at determining the main determinants of potential changes in dietary habits. The authors examined changes in the patterns of beef consumption in Brazil, Spain, and Turkey among 412, 407 and 424 respondents, respectively. A self-administered questionnaire was employed. The results indicate that economics and accessibility were the most important factors that changed consumer behaviour and decreased beef consumption, mostly among Brazilian and Turkish respondents.
Dear reviewer, the authors would like to thank you for your opinion and suggestions. Kindly be informed that the modifications, in the manuscript, are highlighted in yellow.
The topic is highly relevant for various disciplines and has numerous policy implications. However, the paper has several serious shortcomings which need to be addressed:
- The main issue is the readability of the paper. Although grammatically, sentences and paragraphs are, for a large part, appropriate, they are often difficult to understand what the authors wanted to comminucte. Paragraphs are often too short, not following the scientific paragraph structure, so the main idea of a paragraph is often not developed. The paragraphs are sometimes only loosely linked together, jumping from one topic to another. This makes the job of a reader very difficult.
The Introduction topic's (Line: 36-113) entire structure was revised, and the paragraphs were rearranged for improved idea coherence and a more fluid reading. New references were also incorporated to help with the development of the study themes. Your recommendation to enhance the paragraph structure and linking was adopted throughout the entire manuscript.
Just one example from the first paragraph:
“Although the relevance of the selection criteria for beef can vary over time and 35 among nations, they are generally the same.” Selection criteria for whom?
In this sentence the “selection criteria” refers to the criteria of the consumers. However, this sentence was changed, since it was not understandable.
(Line 48-49)
And the second paragraph:
“Assessing the quality of beef can be more challenging than assessing the quality of 40 other consumer products.” Assessed by who? Second, is it more difficult than, for example, pork? If so, why?
That sentence was withdrawn.
- Besides general readability, there are grammatical issues as well, for example:
“The con- 26 sumers whose purchasing intentions were most significantly influenced by confidence factors were 27 Brazilians e Turkish, and to a minor extent, Spanish consumers. Lifestyle has been demonstrated to 28 alter consumption patterns,”
“Brazilians e Turkish” needs correction.
The manuscript has been reviewed by a native English speaker, according to your request.
What are confidence factors? When mentioned for the first time like this, the reader won’t understand it.
The authors are in agreement in modifying the word confidence by credence (those product attributes that are difficult to assess even after consumption) in the manuscript. In the Abstract, when mentioning the credence factors, some examples of these factors were added.
(Line 28-29)
“Lifestyle has been demonstrated to 28 alter consumption patterns, and, therefore, must be carefully considered by the industry, taking into 29 account cultural differences and consumer needs.” What about lifestyle specifically? Unclear.
The Abstract section has also been updated with examples of lifestyle factors.
(Line 30-31)
“According to the studied population, eating beef 30 has no negative impact on one's health.”
This is again unclear. If the authors wanted to says that the perception of respondents is that beef consumption has no impact on health, I suggest rephrasing it that way. Additionally, this needs to be followed with a sentence on the negative impact of red meat, in general, and beef, specifically, on health outcomes, with, preferably, three or more references confirming this link, which is well-established in the literature.
We concur that the term mentioned is definitely imprecise, however the findings show no negative health impacts. It just concerns the non-impact on the study's participants' health; it does not address the other populations. The revised version is as follows: This statement has been modified.
(Line 32-33)
- “The sociodemographic characteristics of the region, such as food preferences, life- 48 style, the accessibility of products, traditions, and even the seasons, can all have an impact 49 on consumption and purchase habits [11-15]. 50”
This is true, but not further explained in any way. The theoretical part needs a Literature review on sociodemographic and other determinants of meat consumption.
Information about socioeconomic aspects was incorporated into the theoretical part of the manuscript.
(Line 92-104)
- “The world's consumption of beef has been increasing as a result of the population's 51 rising food requirements. Economic expansion has led to this demand, which is also a 52 consequence of other structural changes including population growth and the urbaniza- 53 tion process, particularly in developing nations [16]. 54”
In what way do economic expansion and urbanization drive meat consumption?
Above all, beef is a costly product, economic expansion and structural changes, such as urbanization, especially in developing countries, favor access and monetary power to meat consumption. For clarity, the following sentence has been changed.
(Line 37-42)
- “Health care has been a topic of significant discussion, and this aspect may have con- 72 tributed to changes in consumption patterns. On the one hand, beef consumption is re- 73 lated to the fact that it provides great health benefits, such as the provision of iron and 74 crucial nutrients. On the other hand, researchers suggest that a high intake of beef has 75 been associated with chronic diseases such as diabetes, cancers, cardiovascular disease, 76 and high mortality rates. Because beef is both considered healthy and unhealthy at the 77 same time, this ambiguity creates conflict in the consumer's mind [23-26]. 78”
This is, unfortunately, very misleading for the reader. What reputable scientific research or public health institutions claim, based on empirical evidence, that increased beef consumption is healthy compared to less beef consumption? If any, a reference is needed. Second, “provision of iron and crucial nutrients” is insufficient to claim a product has beneficial health effects. Many products have iron and crucial nutrients, but the health impact of a particular food or food group involves whether these nutrients can be acquired in other products, which tend to have (more) beneficial (or less detrimental) health consequences. Third, as iron is mentioned, let me note that high iron stores have been found to be associated with an increased risk of metabolic syndrome, diabetes, cardiovascular diseases and some forms of cancer. Even in the case of greater bioavailability of animal-derived iron (heme-iron), it has been associated with adverse health outcomes compared to plant-based non-heme iron.
To sum up, authors should put more thought into the construction of their sentences when making health claims of specific products/food and align them with scientific evidence.
You are absolutely right, an increase in beef consumption is not indicative of an increase in health, so, we eliminated this information. As we have indicated, in response to your first suggestion, we have improved the information provided in the manuscript by reorganizing it and including more scientific evidence.
(Line 84-91)
- “Research has been showing how, in comparison to past periods, today's meat pro- 86 duction systems and the problems associated with them are given more consideration 87 than they were [15,18].” Unclear.
Furthermore, the present-day meat production system is getting more mass-based, extensive and deleterious for the planet and health (and animals) than it was 50 years ago. The authors themselves state that meat consumption is on the rise globally.
The paragraph has been changed to make the information more understandable. (Line 75-83)
We understand that the debate surrounding the role of meat as part of a healthy and sustainable diet is a contentious and evolving issue. Although animal-sourced foods can have a negative impact on human health, the environment and animal welfare, and plant-based diets can be nutritionally satisfying and are preferred by a significant portion of the population, the majority of human subjects are omnivores and meat consumption has many positive impacts.
Besides that, the health benefits of meat can be enhanced, animal welfare standards can be improved, and negative health and environmental impacts can be mitigated through behavioral change and technological innovations. We believe that the role of meat in the diet needs to be carefully considered in each context and within local and regional realities and emotive and conflicting messages should be avoided wherever possible to allow informed consumer decision making.
- “The application of a self-administered questionnaire to a total of 1243 regular con- 114 sumers of beef in Brazil, Spain and Turkey was performed, for this study, during the sec- 115 ond semester of 2021. 116”
The sample description is too vague. It is not clear how representative (and representative of whom) the sample is in each of the three countries.
The exclusive sample of "regular consumers of beef" was the selection criterion for the study. The population of each country analyzed is not reflected in the samples for this study. The subtopic 2.1. Sample - Consumers and Study Location of Materials and Methods, provides a better description of this information.
(Line 117-118)
“The questionnaire, which was created with the help of the Google.forms software 137 (Web Application - Google Platform), was sent online to consumers in the native language 138 of each country: Portuguese for Brazil, Spanish for Spain, and Turkish for Turkey. 139 Data were collected using two non-probabilistic sampling techniques: conventional 140 sampling, in which subjects are chosen for their accessibility, and chain sampling, which 141 is used to categorize subjects with specified criteria, in this case, beef consumers [41,42].”
It is unclear how, specifically, data was collected with regard to the conventional sample (“subjects were chosen” – but according to what criteria? What was the response rate? The same goes for chain sampling.). All this makes the reproducibility of the study practically impossible.
Information about conventional sample and chain sampling methods was added to topic 2.1. Sample - Consumers and Study Location of Materials and Methods (this last nomenclature method was replaced by the synonym snowball sampling). Additionally, we specified why the number of responses we required and who was responsible for delivering the surveys.
(Line 125-132)
- “Aspects of confidence as a decisive factor in the purchase. “Who’s confidence in what?
Despite the fact that both terms are used in the literature, we chose the more didactic term "credence" in place of the word "confidence." Examples and a definition of credibility factors have been added to the text.
(Line 68-70)
- “The influence of lifestyles in beef consumption pattern” “In” à on. Pattern à Patterns.
Indeed, the grammar would need to be corrected, however, this sentence has been removed from the manuscript. The points that described the themes that were investigated have been revised in light of reviewer 2's suggestion that we present the questionnaire's questions directly.
(Line 138-149)
- “In summary, contrasted to Turkey, Brazil and Spain surveyed more female in- 170 dividuals. Younger consumers, those between the ages of 18 and 34, were more prevalent 171 across all countries.”
ANOVA mean differences may primarily be due to the differences in sample characteristics among the three countries. This puts the validity of the results under serious doubt.
- “it should be highlighted that this sample is not representative of the population as 175 a whole.”
The “national” results may primarily be due to sample characteristics, while the readers don’t know much about data collection. Sociodemographic differences between samples should be taken into account (empirically controlled for), and then mean differences can be discussed.
The statistical method used in this study, although simple, offers very reliable and consistent results with a relatively large sample size used to compare variances between medians (or means) of different groups (that we selected as countries). Other methods could be used, such as Latent Class Analysis or Cluster, to obtain consumer segments. However, our objective was not to define the characteristics that identify or estimate the prevalence of groups and classify each individual into groups.
- The Results section: “The economic aspect is one of the classic factors that explain changes in consumption 184 habits [2,3,19].” Stating references need to be done in the Theoretical part and then in the Discussion section.
We agree, the phrase was moved to the Discussion part.
- “In this study it is observed how an economic crisis can have a great impact on the 215 consumption of beef, however, the economic factor has been for some years a factor of 216 decrease in consumption as exemplified by results of a 2016 study with consumers of beef 217 from Brazil and Spain found that the majority of the survey participants, who were fre- 218 quent consumers of beef, had not changed their consumption levels in the previous year’s 219 [32]. 220”
What is the specific evidence/measure that indicates that the economic crisis had a great impact on the consumption of beef?
In this context, the expression economic crisis cannot be used as a justification. As a result of your observation, the term "crisis" has been dropped.
(Line 217-219)
- Table 4:
“There was no introduction of other protein sources intended to replace beef”
“Less beef is consumed due to the introduction of other protein sources.”
“More beef is consumed, even with the introduction of other protein sources.”
The first statement does not necessarily indicate that there was no change in meat consumption but merely that reduced meat was not replaced with other sources of protein.
It cannot be said that beef intake was decreased, but this phrase relates to the fact that other proteins were not introduced into the diet to substitute beef. In case something isn't clear, please let us know.
- “Table 5. Introduction of other sources of protein to replace beef in the Brazilian, Spanish and Turkish 253 population questioned.” What is the number of respondents in each country in this particular analysis? Not the full sample surveyed?
This question, "Introduction of other sources of protein to replace beef in the Brazilian, Spanish, and Turkish population questioned," was only destined for consumers who responded to the prior questioning that had begun consuming less beef as a result of the addition of other sources of protein to their diet. We made a mistake by not making clearer the requirements for this. The sampled number has also been corrected.
(Line 269-271 and 275, 276)
- 3.5 “Lifestyle habits, which are closely connected to socioeconomic and demographic fac- 351 tors, can change how much beef is consumed [3,10,55]. 352”
A sociodemographic literature review on beef and meat consumption needs to be added to the theoretical part. The authors may take advantage of MDPI journals’ and other publications on determinants of food choice and include them for an interested reader, e.g.,:
1) https://doi.org/10.3390/su132313036
2) Gossard, M.H.; York, R. Social Structural Influences on Meat Consumption.Hum. Ecol. Rev.2003,10, 1–9.
3) Koch, F.; Heuer, T.; Krems, C.; Claupein, E. Meat consumers and non-meat consumers in Germany: A characterisation based on results of the German National Nutrition Survey II.J. Nutr. Sci.2019,8, e21
The references are welcomed, and we are interested in using some of them.
New references have been cited in the manuscript:
- Leroy, F.; Barnard, N. D. Children and adults should avoid consuming animal products to reduce risk for chronic disease: NO. American Journal of Clinical Nutrition 2020, 112 (4), 931-936.
- Barnard, N. D.; Leroy, F. Children and adults should avoid consuming animal products to reduce risk for chronic disease: YES. American Journal of Clinical Nutrition 2020, 112 (4), 926-930.
- Bonnet, C.; Bouamra-Mechemache, Z.; Requillart, V.; Treich, N. Viewpoint: Regulating meat consumption to improve health, the environment and animal welfare. Food Policy 2020, 97.
- De Boer, J.; Schosler, H.; Aiking, H. "Meatless days" or "less but better"? Exploring strategies to adapt Western meat consumption to health and sustainability challenges. Appetite 2014, 76, 120-128.
- Ge, J. Q.; Scalco, A.; Craig, T. Social Influence and Meat-Eating Behaviour. Sustainability 2022, 14 (13).
- Kirbis, A.; Lamot, M.; Javornik, M. The Role of Education in Sustainable Dietary Patterns in Slovenia. Sustainability 2021, 13 (23).
- Britwum, K.; Bernard, J. C.; Albrecht, S. E. Does importance influence confidence in organic food attributes? Food Quality and Preferenc 2021, 87.
- Koch, F.; Heuer, T.; Krems, C.; Claupein, E. Meat consumers and non-meat consumers in Germany: a characterisation based on results of the German National Nutrition Survey II. Journal of Nutritional Science 2019, 8.
- Li, X. G.; Jensen, K. L.; Clark, C. D.; Lambert, D. M. Consumer willingness to pay for, beef grown using climate friendly production practices. Food Policy 2016, 64, 93-106.
- Malek, L.; Umberger, W. J. Distinguishing meat reducers from unrestricted omnivores, vegetarians and vegans: A comprehensive comparison of Australian consumers. Food Quality and Preference 2021, 88.
- Rieger, J.; Kuhlgatz, C.; Anders, S. Food scandals, media attention and habit persistence among desensitised meat consumers. Food Policy 2016, 64, 82-92.
- “This study attempted to comprehend how beef consumption patterns among Brazil- 397 ians, Spanish and Turks consumers have changed over time” What was the time period asked about?
The last two years were the time frame specified. The phrase has been revised. Information about socioeconomic aspects was incorporated into the theoretical part of the manuscript.
(Line 463)
- “The results also showed that the behavior of beef consumers was the preference for 406 replacing beef with other animal sources of protein.” Unlcear.
We agree that the statement needs to be clarified, thus it has been revised.
(Line 471-473)
- There is no Discussion section that would put results into context with broader literature. The Limitations section/paragraph is missing.
In order for the findings to be supported by the literature, scientific references were added, and a paragraph indicating the study's limitations (Line 454-460) was added.
I suggest the authors thoroughly revise the paper, as the topic of the paper is of the most importance to humans, animals and the planet.
We sincerely appreciate your willingness to contribute to improving our manuscript.

Author Response
This study focusses on Changes in the current patterns of beef consumption and consumer behavior trends among Brazil, Spain, and Turkey. The topic is interesting, and major changes are needed for publishing on Foods:
Dear reviewer, the authors would like to thank you for your opinion and suggestions. Kindly be informed that the modifications, in the manuscript, are highlighted in yellow. The manuscript has been reviewed by a native English speaker, according to your request.
- I think the writing style need change in the whole article. You do not have to make a sentence into a paragraph. There are so many paragraphs in the article. I do not think it is good for reading.
The Introduction topic's (Line: 36-113) entire structure was revised, and the paragraphs were rearranged for improved idea coherence and a more fluid reading. New references were also incorporated to help with the development of the study themes. Your recommendation to enhance the paragraph structure and linking was adopted throughout the entire manuscript.
New references have been cited in the manuscript:
- Leroy, F.; Barnard, N. D. Children and adults should avoid consuming animal products to reduce risk for chronic disease: NO. American Journal of Clinical Nutrition 2020, 112 (4), 931-936.
- Barnard, N. D.; Leroy, F. Children and adults should avoid consuming animal products to reduce risk for chronic disease: YES. American Journal of Clinical Nutrition 2020, 112 (4), 926-930.
- Bonnet, C.; Bouamra-Mechemache, Z.; Requillart, V.; Treich, N. Viewpoint: Regulating meat consumption to improve health, the environment and animal welfare. Food Policy 2020, 97.
- De Boer, J.; Schosler, H.; Aiking, H. "Meatless days" or "less but better"? Exploring strategies to adapt Western meat consumption to health and sustainability challenges. Appetite 2014, 76, 120-128.
- Ge, J. Q.; Scalco, A.; Craig, T. Social Influence and Meat-Eating Behaviour. Sustainability 2022, 14 (13).
- Kirbis, A.; Lamot, M.; Javornik, M. The Role of Education in Sustainable Dietary Patterns in Slovenia. Sustainability 2021, 13 (23).
- Britwum, K.; Bernard, J. C.; Albrecht, S. E. Does importance influence confidence in organic food attributes? Food Quality and Preferenc 2021, 87.
- Koch, F.; Heuer, T.; Krems, C.; Claupein, E. Meat consumers and non-meat consumers in Germany: a characterisation based on results of the German National Nutrition Survey II. Journal of Nutritional Science 2019, 8.
- Li, X. G.; Jensen, K. L.; Clark, C. D.; Lambert, D. M. Consumer willingness to pay for, beef grown using climate friendly production practices. Food Policy 2016, 64, 93-106.
- Malek, L.; Umberger, W. J. Distinguishing meat reducers from unrestricted omnivores, vegetarians and vegans: A comprehensive comparison of Australian consumers. Food Quality and Preference 2021, 88.
- Rieger, J.; Kuhlgatz, C.; Anders, S. Food scandals, media attention and habit persistence among desensitised meat consumers. Food Policy 2016, 64, 82-92.
- Line 113: Did you design some question to test the availability and reliability of the questionnaire survey.
A pilot test of the questionnaire was conducted before to its distribution, but since there were only a few questions, its authors did not see the need to include a question to evaluate its reliability. However, the risk of response dishonesty exists when using a self-administered online questionnaire. (This information was added to a paragraph designed to highlight the study's limitations). On the lines (454-460), you may find the paragraph describing the limitations of the study that were added at the request of reviewer 2.
The methodology adopted, conventional sampling and snowball sampling, "conventional sampling, in which individuals were chosen for their accessibility, and snowball sampling, which was used to access with specified criteria, in this case beef consumers [41,42]." Line (123-125) provided better efficiency and availability in answering the questionnaire in order to prevent other biases.
Additionally, was tried made clear in the consent form, which was made available immediately as the consumer opened the questionnaire, that answering the questionnaire was a voluntary act, that it would take them about a few minutes to complete it, and that they had the option to leave if they found it to be uncomfortable or inconvenient in any other way.
3.Line 136: So, all the 1243 regular consumers did the investigation well? Were there some people did not finish the survey? Was there no vegetarian?
To address the questions highlighted, information has been added and restructured under the Materials and Methods subtopic 2.1. Sample - Consumers and study location.
Line (117-118) “collecting a total of 1243 complete and valid responses from regular beef consumers,”
4.Line 144: What are the six closed-ended questions? Please write them directly.
Taking into account your suggestion, the six closed-ended questions were described directly.
Line (138-149)
- Has your beef purchasing pattern changed due to economic reasons in the last two years?
- Have you started using other sources of protein in your diet with the intention of replacing beef in the last two years?
- Which protein sources below have you consumed to replace beef in your diet in the last two years?
- Have any of these factors (credence) affected your level of credence and, as a consequence, the beef purchase intention?
- Has your beef consumption changed because of those health-related factors? (The factors were described below the question).
- Due to the following (lifestyle) factors: Have you increased, decreased, or not changed your beef consumption?
5.Line 156: Why you choose Pearson’s chi-square and z-test to analyze your stata. Did you have some references?
The statistical method used in this study as the Chi-square test and z-test, despite being simple, offers fairly reliable and consistent results with a relatively large sample size (n> 1000 cases). This methodology was used to verify the statistical significance of the association observed in a cross-tabulation, assuming that the null theory is accepted.
We used this data analysis based on the reference of our recent paper published in the Foods Journal: Magalhaes, D. R.; Campo, M. D.; Maza, M. T. Knowledge, utility, and preferences for beef label traceability information: A cross-cultural market analysis comparing Spain and Brazil. Foods 2021, 10(2).
6.Line 396: Please simply your Conclusion.
We sincerely appreciate your recommendations and would like to know more about what information you consider could be simplified or removed from the conclusion.

Round 2
Reviewer 1 Report
The paper has been sufficiently revised and is now much improved.